# Investigation of Electromagnetic Scattering Mechanisms from Dynamic Oil Spill–Covered Sea Surface

**Dongfang Li [1,2], Zhiqin Zhao [2,\*], Wenying Ma [1] and Yajuan Xue [1]**

[1] The School of Communication Engineering, Chengdu University of Information Technology, Chengdu 610225, China; dongfang@cuit.edu.cn (D.L.); mwy@cuit.edu.cn (W.M.); xueyj0869@cuit.edu.cn (Y.X.)

[2] The School of Electronic Science and Engineering, University of Electronic Science and Technology of China, Chengdu 611731, China

\* Correspondence: zqzhao@uestc.edu.cn; Tel.: +86-028-6183-0877

**Abstract:** The electromagnetic (EM) scattering mechanism of dynamic oil spill–covered sea surface area is studied in this manuscript. Utilizing the theory of oil film diffusion combined with oil spill volume, a three–dimensional (3D) geometric model of dynamic oil spill–covered sea surface area is established. The changes of the geometric structure and statistical characteristics of the sea surface area under the influence of the oil film are also analyzed. The thinner the oil spill thickness, the more sensitive it is to the sea surface slope and wave height. The facet-based hybrid model and the multilayer dielectric scattering method are combined to measure the EM scattering on the sea surface when covered by dynamic oil spills. In addition, the hydrodynamic and tilt effect are discussed. The EM scattering mechanism of the dynamic oil spill–covered sea surface area is revealed, and the tilt modulation is greater than the hydrodynamic effect in the dynamic process of an oil spill. It provides an important reference for the remote sensing monitoring of oil.

**Keywords:** electromagnetic scattering; oil spill; oil diffusion; remote sensing; sea surface

## 1. Introduction

Oil spills pose a great threat to the marine economy and environment. It is of great importance to quickly and accurately monitor oil spill situations and extract oil spill information for disaster assessment and cleanup after oil spills. Microwave remote sensing technology, especially the synthetic aperture radar (SAR), is an effective method for detecting oil spills in the ocean. The SAR data can better understood by analyzing the investigation of EM scattering of oil spill surface [1–3].

The fine geometric model is used as the basis for understanding EM scattering from the sea surface when covered by dynamic oil spills. There are many methods to describe wave height, with the most commonly used generating tool being sea spectrum. Pierson and Moscowitz developed the famous Pierson–Moskowitz (PM) sea spectrum—also known as the gravity spectrum (which can describe fully grown wind-driven waves)—after the spectral analysis of ocean observation records in the North Atlantic Ocean [4]. Hasselman et al. [5] carried out a large number of observations and collected statistical data in regions off the west coasts of Germany and Denmark. The JONSWAP sea spectrum model was obtained, and the wind zone factor was added to the spectrum, which can describe different growth states of the waves. Elfouhaily et al. [6] proposed the full wavenumber spectrum—the Elfouhaily sea spectrum—based on the pooling of laboratory observation data. This spectrum modifies and combines multiple sea spectra, such as the PM and JONSWAP spectra. When oil spills occur, it affects the sea spectrum and surface roughness. Some scholars have proposed a suppression ratio, such as the Lombardini model, Jenkins model, Ermakov model, and Nicolas Pinel model [7–10] to describe how the sea spectrum generation is affected by oil spills. These viscous damping models can

effectively describe the viscous loss brought on by the Marangoni effect. The above geometric of the sea surface generated by the sea spectrum method does not consider the dynamic changes of the sea surface during an oil spill. This dynamic change means that when the oil spill occurs, the film spreads around rapidly, forming a thin spillover layer that covers the sea surface. In the process of diffusion, how oil spill diffuses and how the oil spill thickness changes should follow a certain diffusion theory [11–13]. Therefore, it is meaningful to establish the geometric model of a three-dimensional (3D) dynamic sea surface area covered with an oil spill, which provides the basis for the subsequent study of EM scattering.

There are many analytical and numerical methods to evaluate EM scattering from the sea surface covered by a dynamic oil spill. The numerical method has high accuracy, especially at large incidence angles, while it has low computational efficiency to orient the sea surface with its electrically large size, such as multilevel fast multipole algorithm (MLFMA) [14], etc. The analytical method has obvious advantages in efficiency and accuracy when small and medium incidence angles are considered, such as using the two-scale model (TSM) [15–19], integral equation method (IEM) [20], small slope approximation (SSA) [21,22], etc. Many authors have adopted different methods to evaluate the EM scattering from the sea surface covered by oil spills. Marina Mityagina [23] used the geometric optical (GO) approximation method [15] to estimate the EM scattering influence affected by different oil spill thicknesses. The multiple reflections in the oil spill were considered, and the Fresnel reflection coefficient was modified. Ayari [24] adopted the TSM, and Guo et al. [25] proposed the improved SSA-2 to evaluate the EM scattering of the sea surface covered by oil spills—the simulation results were compared with the numerical method. Zheng et al. [26] were devoted to investigating the EM backscattering of oil–free and oil spill–covered sea surfaces in different wavebands using the SSA-2 method. The calculation efficiency and accuracy of the above methods need to be improved. A large number of measured data prove that oil spills have obvious effects on backscattering of the sea surface. In the range of low and medium incidence angles, the presence of an oil spill will reduce the radar backscattered echo power [27–32]. This suggests that we need to study what the main mechanism at work is in oil diffusion.

In calculation of the EM scattering from the sea surface covered by a dynamic oil spill, the modified facet–based two–scale method (MFTSM) [19] and the multilayer dielectric scattering model [20,33] are adopted, which indicates the variation of EM scattering by dynamic diffusion of the oil spill. The main contribution of the manuscript is to study the EM scattering mechanisms from dynamic oil spill–covered areas on the sea surface. It provides a new theoretical reference for predicting oil leakage amounts.

This manuscript is organized as follows: In part 2, the 3D geometric model of the sea surface covered by dynamic oil spills is established. This model describes the dynamic changes of the sea surface from clean surface, to oil spill, to oil diffusion. Part 3 describes the EM scattering model. The normalized RCS (NRCS) coefficient is obtained and compared with measured data. The specific effects of hydrodynamic modulation and tilt modulation are discussed. In addition, the NRCS versus the oil spill thickness, area of oil coverage, incident angles, frequencies, and wind speeds are discussed in part 4. Part 5 is the conclusion.

## 2. The 3D Geometric Model of Oil Spill–Covered Sea Surface Area

### 2.1. Sea Spectrum

The 3D geometric model of the sea surface is usually realized using the linear filtering method. When an oil spill occurs, oil film will inhibit the sea waves, especially the capillary–gravity waves, making the sea waves relatively smooth. The relationship between the clean sea spectrum and oil sea spectrum [2,34] is:

$$S_{oil} = \frac{S_{clean}}{y_s(K)} \tag{1}$$

where $k$ is the spatial wavenumber of sea waves, and $S_{clean}$ is the oil–free sea spectrum. In this manuscript, the Elfouhaily [6,19] spectrum is adopted to generate the clean sea. $y_s$ is the attenuation coefficient, which describes the attenuation effect of sea waves affected by oil film. Viscous damping is the most direct and important influencing factor of oil spills on sea waves. When a certain amount of oil spill covers the sea surface, the oil spill, and sea surface can be regarded as a bilayer fluid model. In this case, the viscous damping of the oil spill should consider oil film thickness, the surface tension, elasticity, viscosity, and other parameters of sea water and oil spill. The relative attenuation coefficient $y_L$ of the Jenkins and Jacobs model [8] can be expressed as:

$$y_L(k) = \frac{\mathrm{Re}(\delta)}{2v}, \tag{2}$$

$$\delta = \frac{\left\{ \begin{array}{c} 2v + 0.5v_T + j\Gamma^{-0.5}[\gamma(1 - \rho_+) - \gamma_-]D + \\ \frac{1}{2v^{0.5}j^{0.5}}\rho + D\Gamma^{0.25}[v_T + j\rho + D\Gamma^{0.5}(R^2 - 1)] \end{array} \right\}}{\left( 1 + \frac{1}{v^{0.5}\Gamma^{0.25}}j^{0.5}v_T + \frac{1}{j^{0.5}v^{0.5}}\rho + D\Gamma^{0.25} \right)}, \tag{3}$$

$$v_T = \frac{\chi_+ + \chi_-}{n} + v_{s+} + v_{s-} + 4\rho_+v_+D + \frac{Dv_{E+}v_{E-}}{\rho_+v_+}, \tag{4}$$

$$\begin{cases} R = \frac{\rho_+ + \gamma_\pm}{\rho + \Gamma}, \; n = -j\sqrt{\Gamma}, \; \Gamma = 1 + \gamma, \; \gamma = \gamma_+ + \gamma_- \\ v_{E\pm} = \frac{\chi_\pm}{n} + v_{s\pm}, \; j = \sqrt{-1} \end{cases}, \tag{5}$$

where $\gamma$ is the tension of sea water, and $D$ is the oil thickness. The other specific relevant parameters of oil spill are obtained in Table 1 in this manuscript.

**Table 1.** Relevant parameters of oil spill (see [8]).

| Physical Parameters | Values |
|---|---|
| Water density ($\rho$) | $1023 \text{ kg} \cdot \text{m}^{-3}$ |
| Oil density ($\rho_+$) | $900 \text{ kg} \cdot \text{m}^{-3}$ |
| Surface viscosity ($v_{s+}$) | 0 |
| Interfacial viscosity ($v_{s-}$) | 0 |
| Kinematic viscosity of water($v$) | $10^{-6} \text{ m}^2 \cdot \text{S}^{-1}$ |
| Kinematic viscosity of oil ($v_+$) | $10^{-4} \text{ m}^2 \cdot \text{S}^{-1}$ |
| Surface elasticity ($\chi+$) | $15 \text{ mN} \cdot \text{m}^{-1}$ |
| Interfacial elasticity ($\chi-$) | $10 \text{ mN} \cdot \text{m}^{-1}$ |
| Surface tension ($\gamma+$) | $25 \text{ mN} \cdot \text{m}^{-1}$ |
| Interfacial tension ($\gamma-$) | $15 \text{ mN} \cdot \text{m}^{-1}$ |

Oil leakage is a dynamic diffusion process. Discussing the state from the initial thick oil spill with low oil coverage to the thin oil spill with large oil coverage, the concept of oil spill coverage needs to be introduced. According to the Marangoni damping theory, the attenuation function including both the oil spill thickness and the oil spill coverage rate [7] can be expressed as:

$$y_S(K) = [1 - F + F/y_L(K)]^{-1}, \tag{6}$$

where $F$ is the oil spill coverage rate; and when $F = 1$, $y_s(k) = y_L(k)$, it represents the oil film coverage of the whole observed sea surface in this situation.

Figure 1 is the density function comparison between the clean sea spectrum and oil film sea spectra at various wind speeds. It shows the distribution of the oil spill sea spectrum as a function of the spatial wavenumber. The abscissa represents the spatial

wavenumber, and the ordinate represents the spectral energy density. $K_{cut}$ refers to the cut-off wavenumber. The wavenumbers between $K_{min}$ and $K_{cut}$ indicate that these wave numbers generate large scale waves, $K_{Bragg}$ means Bragg wavenumber, the capillary wave is reduced to cosine wave generated by Bragg wavelength.

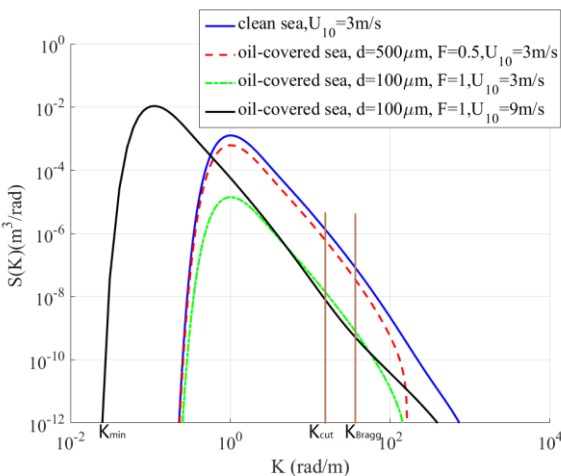

**Figure 1.** Comparison between clean sea spectrum and oil film sea spectra.

With the dynamic diffusion of an oil spill, the energy of the sea spectrum changes. As shown in Figure 1, at the beginning of the oil spill, the coverage of oil film is low, and the oil spill is thick. As time goes on, an increasingly larger area of the sea surface is completely covered by the oil spill, and the oil spill thickness becomes thinner. In the whole dynamic change of an oil spill, in comparison to clean sea spectrum, the spectra peak decreases with the decrease of oil spill thickness and the increase of oil spill coverage. The oil spill reduces the root mean square (RMS) wave height of the sea surface. In addition, in the high–frequency region, the stronger the damping effect brought on by oil film, the more the small–scale waves are affected by the oil film, and the smaller the sea surface slope. Since the energy of the wind-driven sea surface is related to the friction velocity, $U_0 = 0.7\,U_w$ ($U_w$ is the friction velocity of the clean wind-driven sea surface, $w$ is water, and $o$ is oil) is often used to calculate the friction velocity of the sea surface covered by the oil spill [35]. Under the same conditions of an oil spill, the spectra peak moves to the low-frequency region as the wind speed increases. The effect of small–scale waves affected by oil film at high wind speeds is weaker than that at low wind speed; the main reason is that small–scale waves affected by oil film are weaker than the interaction between sea waves.

### 2.2. Oil/Water Mixture

The complex permittivity of crude oil is very small, at about 2 + j0.01 [36]. When the oil spill is emulsified by seawater in the process of diffusion, the oil and water will mix and form a stable emulsion. This phenomenon is called chocolate mousse. At this time, the dielectric constant of oil film will change. Parkhomenko thought the oil/water mixture is composed of a continuous main substance (oil) and water droplets; therefore, the equation obtained by [37] is:

$$\frac{\varepsilon_{mix} - 1}{\varepsilon_{mix} + 2} = \frac{\varepsilon_{oil} - 1}{\varepsilon_{oil} + 2} \cdot P_{oil} + \frac{\varepsilon_w - 1}{\varepsilon_w + 2} \cdot P_w ,\tag{7}$$

where $\varepsilon_{mix}$ is for the mixture of oil and water, $\varepsilon_w$ is for water (in [38], $w$ is water), $\varepsilon_{oil} = 2 + j0.01$, $P_{oil}$ is the oil proportion, and $P_w$ is the water proportion $P_w = (1 - P_{oil})$.

### 2.3. Oil Spill Diffusion

When oil spills into the ocean, it spreads out rapidly to form an oil spill. At the beginning of an oil spill, expansion is the most dominant dynamic process. The expansion of an oil spill is mainly affected by flow field, wind stress, random diffusion, and other

processes such as sedimentation and degradation. With the spread of an oil spill, the oil spill area continues to expand and the oil spill thickness gradually decreases. The study [39] shows that the expansion process is terminated when the final crude oil spill thickness reaches 1 mm and the final gasoline, kerosene, and light diesel oil spill thickness reaches 0.1 mm.

In the oil spill spreading model, Fay [40] first proposed that under a calm sea, oil spills spread in a circular manner, and the spreading range could be measured by the diameter. In an actual oil spill, the sea state has a very important effect on the oil spill area and shape, and wind is one of the important factors. Field monitoring shows that the diffusion of an oil spill in the ocean is not circular but elliptic, and the long axis is consistent with the wind direction [41].

The oil spill ellipse diffusion model [41] is obtained using:

$$S = \frac{\pi}{4}QR, \tag{8}$$

$$Q = C_1[(\rho_w - \rho_0)/\rho_0]^\alpha V^\beta t^\eta, \tag{9}$$

$$R = C_1[(\rho_w - \rho_0)/\rho_0]^\alpha V^\beta t^\eta + C_2 U^\varsigma t^\xi, \tag{10}$$

where $Q$ and $R$ are the major and minor axes of the ellipse, respectively; $U$ is the wind speed (assuming $U = 0$, $\beta = 1/3$, and $\eta = 1/4$); $V$ is the volume of oil spill; $t$ is the time; $P_w$ is the density of sea water; and $P_o$ is the density of the oil spill ($o$ is oil). The other parameters are given by [41].

### 2.4. Sea Surface

The sea surface is made up of multiple cosine waves of different wavelengths and amplitudes. In this manuscript, the sea surface is approximated as large–scale waves superimposed with capillary waves, and capillary waves are reduced to cosine waves generated by Bragg wavelength, which is the main scatterer of radar waves. The advantage of this method is that the large–scale waves are divided by coarse grids, which greatly reduces the number of grids. A coarse grid is used to divide large–scale waves; each coarse grid is distributed with small–scale waves simplified to cosine distribution, and the corresponding wave heights are as follows:

$$z_{large}(x,y,t) = \text{IFFT}_2\left[Z_{large}(K_x,K_y,t)\right] = \text{Re}\left\{\sum_{|K|=K_{\min}}^{|K|\leq K_{cut}} Z_{large}(K_x,K_y,t)\right\}, \tag{11}$$

$$Z_{large}(K_x,K_y,t) = C\left[S(K_x,K_y)\right]^{1/2}\exp\left[j\omega(K_x,K_y)t\right] \cdot R, \tag{12}$$

$$z_{small}(x,y,t) = \text{Re}\left\{\sum_{|K|=K_{Bragg}} Z_{small}(K_x,K_y,t)\right\}, \tag{13}$$

where $j = \sqrt{-1}$; $C$ is a constant; the spatial wavenumber of ocean waves $\mathbf{K} = Kcos\varphi e_x + Ksin\varphi e_y$; $\varphi$ is the wind direction; $S(K_x, K_y)$ is the sea spectrum; $\omega(K_x, K_y)$ is the angular velocity, which is related to the wavenumber $K$; $\omega^2 = gK(1 + K^2/K_m{}^2)$; and $k_m = 363$ rad/m. IFFT2 is a two–dimensional inverse Fourier transform.

In addition, the oil spill will affect the geometric structure and statistical characteristics of the sea surface, such as the contour and surface slope. The RMS wave height and surface slope can be calculated using the following formulas, respectively:

$$\sigma_{RMS\_h} = \sqrt{\int_0^\infty \int_{-\pi}^\pi S(K,\varphi)d\varphi dK}, \tag{14}$$

$$\sigma_{RMS\_s} = \sqrt{\int_0^{\infty} \int_{-\pi}^{\pi} (K \cos \varphi)^2 S(K, \varphi) d\varphi dK,} \tag{15}$$

where $S(K, \varphi)$ is the spectrum.

Figure 2 shows the geometric model changes of large–scale sea surface waves during the process from clean sea surface to oil spill diffusion. In the simulation, $f$ = 3 GHz, $\theta_i = 30°$, $\varphi = 10°$, and $U_{10}$ = 5 m/s. The truncated size of sea surface is 300 m × 300 m.

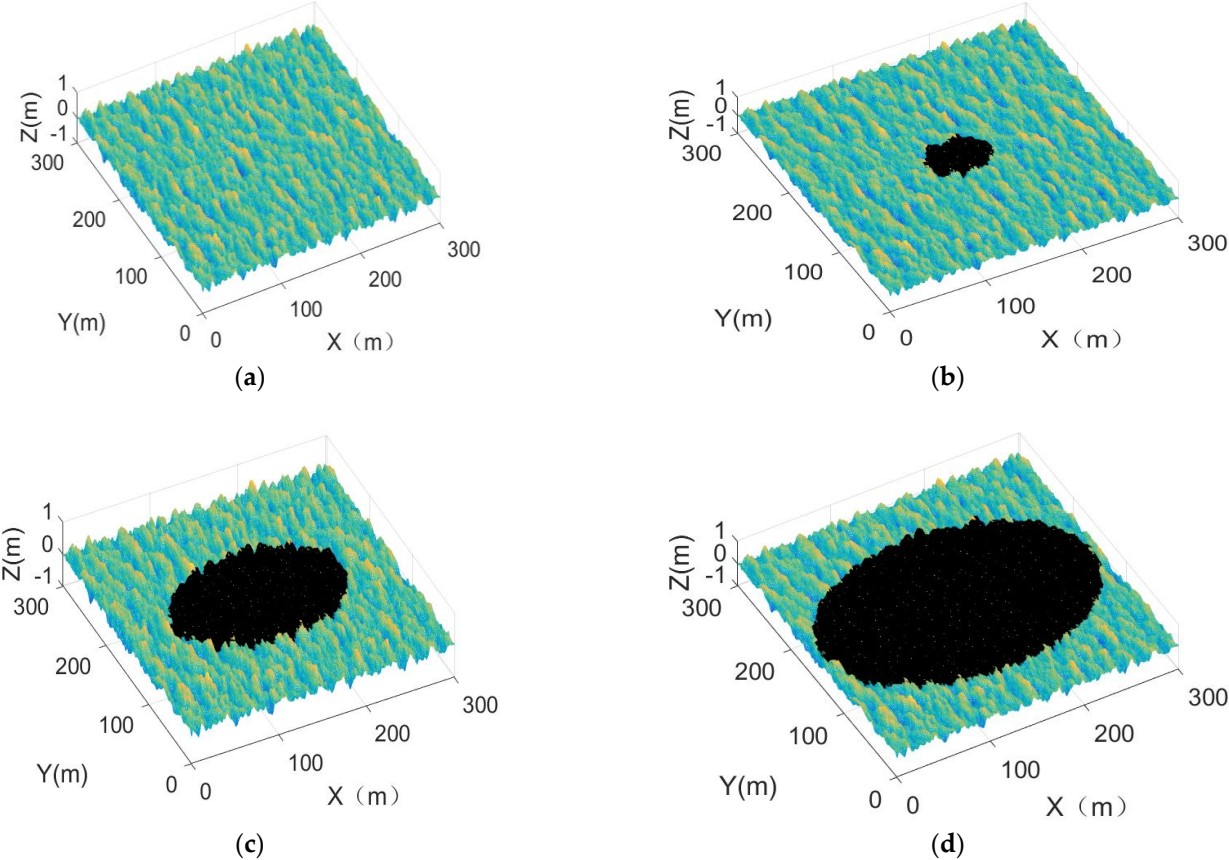

**Figure 2.** The 3D geometric model of oil spill-covered sea surface (the black part shows the oil spill-covered sea area). (**a**) Clean sea; (**b**) Oil spill covers 2% of the surface area; (**c**) Oil spill covers 20% of the surface area; (**d**) Oil spill covers 50% of the surface area.

Figure 2a shows the clean sea surface. Figure 2b shows that oil spill covers 2% of the surface area. The black area indicates oil spill. Figure 2c shows that oil spill covers 20% of the surface area. Oil is slowly spreading in an elliptic pattern. The long axis aligns with wind direction. Figure 2d shows that oil spill covers 50% of the surface area.

Figure 3 shows the geometric features of the oil spill-covered sea surface at $U_{10}$ = 5 m/s. The solid black line represents the clean sea surface; the dashed blue line represents the initial oil spill, which covers 2% of the surface area; and the oil spill thickness is set to 5 mm. The dashed red line shows the stable state of the oil spill, which covers 100% of the surface area (the values are referred to in Refs. [36,39]).

Figure 3a shows the comparison of wave height profiles between clean sea surface and oil spill-covered sea surface. As shown in Figure 3, the wave height in the clean sea is significantly higher than that in the oil spill-covered sea. Figure 3b,c expresses the comparison of sea surface slope. In Figure 3b,c, the slope in the clean sea is obviously larger than that in the oil spill covered. Meanwhile, the slope of the oil spill-covered sea in the y direction is larger than that in the x direction. In addition, when comparing Figure 3a–c, it is revealed that the larger the oil spill that covers the sea surface, the longer the interaction time between the oil film and sea surface, and the larger the influence on

sea surface height and slope. Figure 3c,d displays the comparison of RMS height and slope. The RMS slope is more sensitive than wave height when affected by oil spills. The reason is that the damping effect of an oil spill mainly acts on capillary waves, while large–scale waves mainly determine the RMS height of waves. The RMS slope of ocean waves is more sensitive than the RMS height of the small−scale waves.

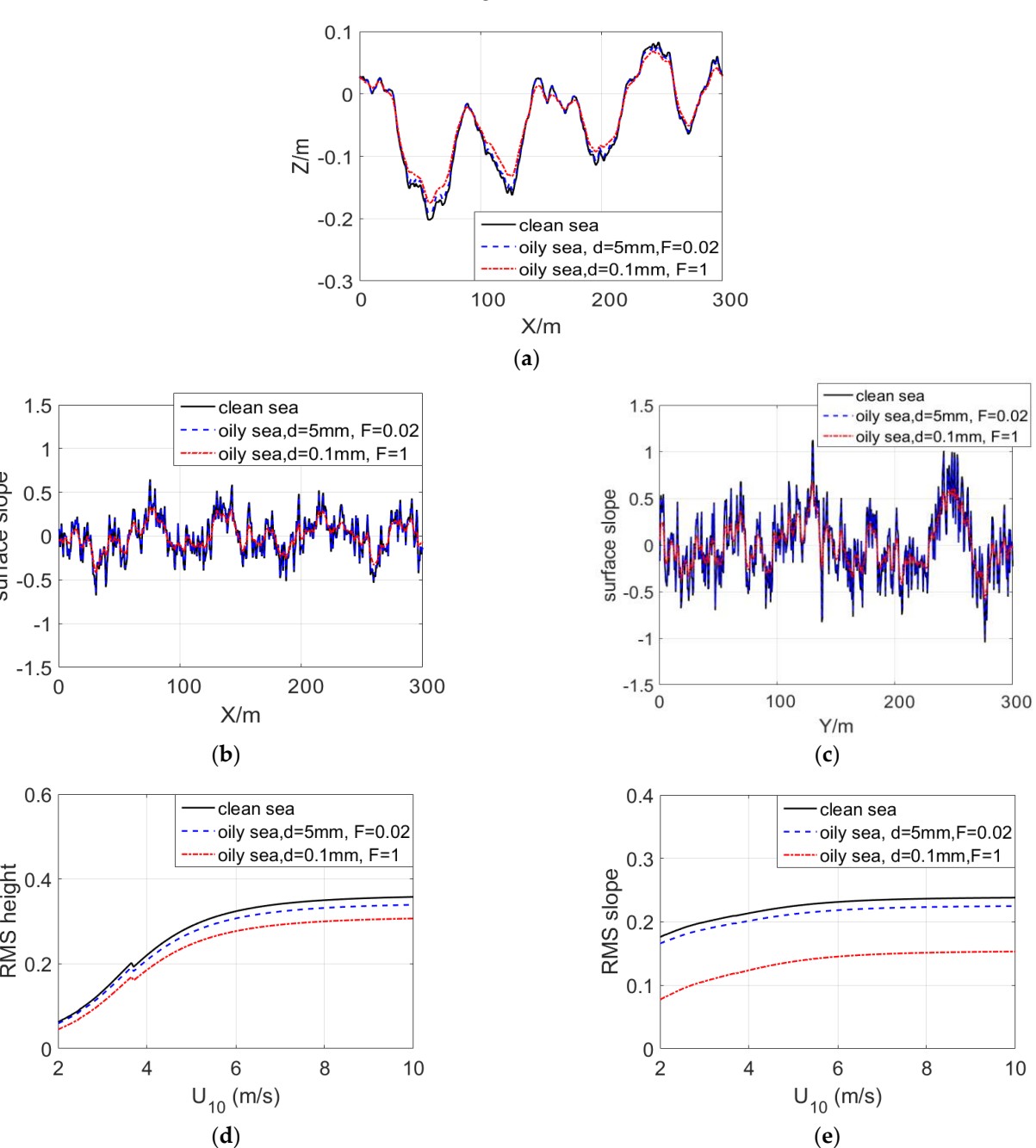

**Figure 3.** The geometric characteristics of oil spill-covered sea. (**a**) Features of sea surface profile along the *x* direction; (**b**) Slope of sea surface profile along the *x* direction; (**c**) Slope of sea surface profile along the *y* direction; (**d**) The RMS height of sea surface; (**e**) The RMS slope of sea surface.

## 3. EM Scattering Model

### 3.1. The EM Scattering of Clean Sea

The sea surface is approximated as large–scale waves superimposed with capillary waves, and the sea surface information represented by small–scale waves can be analyzed using the random EM scattering statistics theory of the sea surface. This combined method

synthesizes the computational efficiency and accuracy problems faced by the whole algorithm. The oil spill–covered sea surface area is covered by oil film layer, which requires a multilayer dielectric scattering model, while the area not covered by oil film can be calculated based on Bragg scattering.

Figure 4 displays the schematic diagram of EM scattering from the sea surface covered by a dynamic oil spill. The sea surface consists of large- and small-scale waves. A coarse grid is used to divide large–scale waves, and each coarse grid is distributed with small-scale waves simplified to cosine distribution. GO is the geometric optics method [15] used to calculate the specular scattering of each clined element, while IEM is the integral equation method [20] used to calculate the diffuse scattering. The global and local coordinate systems are represented by $(x, y, z)$ and $(x', y', z')$. In the global coordinate, $\theta_s$ and $\theta_i$ show the scattering and incident angle. Correspondingly, $\theta_s'$ and $\theta_i'$ are the scattering and incidence angles in local coordinates. $k_i$ and $k_s$ show the incident and scattered wave vector.

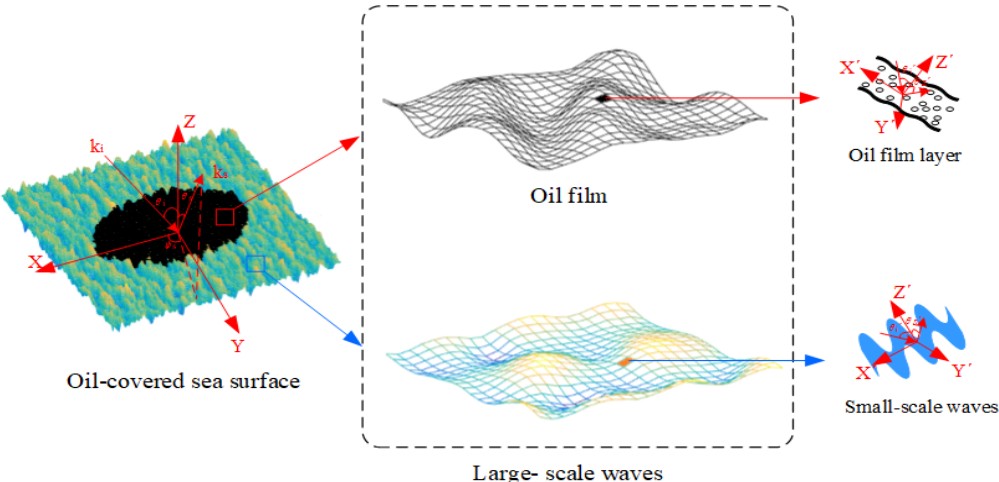

**Figure 4.** Schematic diagram of EM scattering model.

The total scattering field of clean sea is:

$$\mathbf{E}_{PP}^{sea}(\mathbf{k}_i, \mathbf{k}_s) = \frac{e^{jkR}}{2R} \sqrt{\frac{\Delta x \Delta y}{\pi}} \left[ \sigma_{PP}^{GO}(\mathbf{k}_i, \mathbf{k}_s) + \sigma_{PP}^{IEM}(\mathbf{k}_i, \mathbf{k}_s) \right], \tag{16}$$

$$\sigma_{PP}^{GO} = \frac{\pi K^2 q^2}{q_z^4} |U_{PP}|^2 P(Z_x, Z_y), \tag{17}$$

$$\sigma_{PP,mn}^{IEM} = \frac{k^2}{4\pi} \exp\left(-2K^2 \sigma_{mn}^2 \cos^2 \theta_i'\right) \times \sum_{n=1}^{\infty} \frac{|I_{PP}^n|^2}{n!} S_{mn}^{(n')}\left(2K \sin\theta_i', 0\right), \tag{18}$$

where $P$ is for $VV$ or $HH$ polarization, $N' = 1$, $S_{mn}^{(n')}(2K \sin\theta_i', 0)$ corresponds to the Bragg scattering component, $\Delta x$ and $\Delta y$ are sampling intervals along the $x$ and $y$ directions, and $R$ is the straight–line distance between the radar and center of inclined facet element. The specific other parameters are referred to in [15,20].

### 3.2. The EM Scattering of Oil Spill-Covered Sea

In Figure 4, the element of large-scale waves which are covered with an oil spill (black region in Figure 4), the oil spill layer is overlaid on its element. At the beginning, the oil spill is characterized by low oil spill coverage and thick oil spill thickness. Then, due to the influence of other factors, the oil spill begins to spread, the oil spill thickness becomes thinner, and the coverage rate increases. In this case, the multilayer dielectric scattering model [20] is adopted for calculating the EM scattering of oil spill–covered sea area.

Figure 5 represents the four parts of EM scattering contributions of the oil spill layer. The whole EM scattering model of oil film layer [20] is given using:

$$\sigma_{whole}^{oil\_film} = \sigma_{upper} + \sigma_{bspp} + \sigma_{volpp} + \sigma_{inpp}, \tag{19}$$

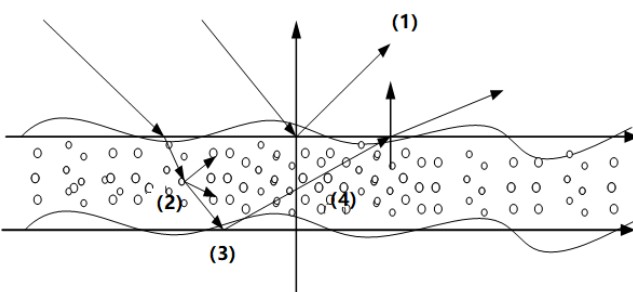

**Figure 5.** The demonstration diagram of EM scattering from oil spill-covered sea; (1), (2), (3), and (4) represent the four part contributions of an oil spill layer.

The total scattering field of oil spill-covered sea is:

$$E_{PP}^{oil\_spill}(\mathbf{k_i}, \mathbf{k}_s) = \frac{e^{jkR}}{2R}\sqrt{\frac{\Delta x \Delta y}{\pi}} \cdot \left[\sigma_{whole}^{oil\_film}(\mathbf{k}_i, \mathbf{k}_s)\right], \tag{20}$$

where $\sigma_{upper}$ is the scattering of upper surface; $\sigma_{volpp}$ is the contribution of second part scattering, which is from the oil spill layer; $\sigma_{bspp}$ is the scattering of bottom surface; and $\sigma_{inpp}$ is the contribution of forth part scattering, which comes from the interaction among the upper surface, oil spill layer, and bottom surface. Relevant expressions of the specific parameters are referred to in [20].

In short, the EM scattering field of the entire sea (including the area covered by and not covered by the oil film) is:

$$E_{PP}^{whole}(\mathbf{k}_i, \mathbf{k}_s) = \sum_{i=1}^{M}\sum_{j=1}^{N}\left((1-c)E_{PP,ij}^{sea}(\mathbf{k}_i, \mathbf{k}_s) + cE_{PP,ij}^{oil\_spill}(\mathbf{k}_i, \mathbf{k}_s)\right), \tag{21}$$

where $c = 0$ or 1 as it shows whether the surface area is covered with oil spill or not.

Therefore, the NRCS of the whole dynamic oil spill-covered sea is:

$$\sigma_{PP}^{whole}(\mathbf{k}_i, \mathbf{k}_s) = \lim_{R\to\infty}\frac{4\pi R^2}{A} \times \left[E_{PP}^{whole}(\mathbf{k}_i, \mathbf{k}_s) \cdot E_{PP}^{whole}(\mathbf{k}_i, \mathbf{k}_s)^*\right]. \tag{22}$$

where $A$ is the area of the sea surface observed.

## 4. Results and Discussion

The proposed model was simulated using the EM scattering model, and the simulation results were analyzed and verified. The experimental data came from the literature [42,43], and the results proved the effectiveness of the proposed model. In addition, the variations of the scattering intensity on the sea surface under different radar parameters, sea state, oil spill thickness, and coverage rate were also analyzed. Finally, the changing trend and influencing mechanisms of EM scattering intensity in the dynamic change process from the initial state to the diffusion of the oil spill are summarized, which provides a theoretical reference for remote sensing monitoring of marine oil pollution.

### 4.1. Comparison of Experimental Data and Simulation Results

Figure 6 displays the comparison of experimental data and simulation results under VV and HH polarizations. The measured data came from the 2010 oil spill, which was measured using the unmanned vehicle synthetic aperture radar (*UAVSAR*) in the Gulf of

Mexico. In Figure 6a, the "dark" region is the oil spill. The two white rectangle areas shown represent oil film and clean sea region. They were chosen to be used as simulation areas, and the results were compared with the measured results and the SSA-II results simulated by Zheng et al. [26]. The simulation results are the statistical average results of 100 samples under the conditions of *VV* and *HH* polarizations. Relevant information about measured data is recorded in Table 2.

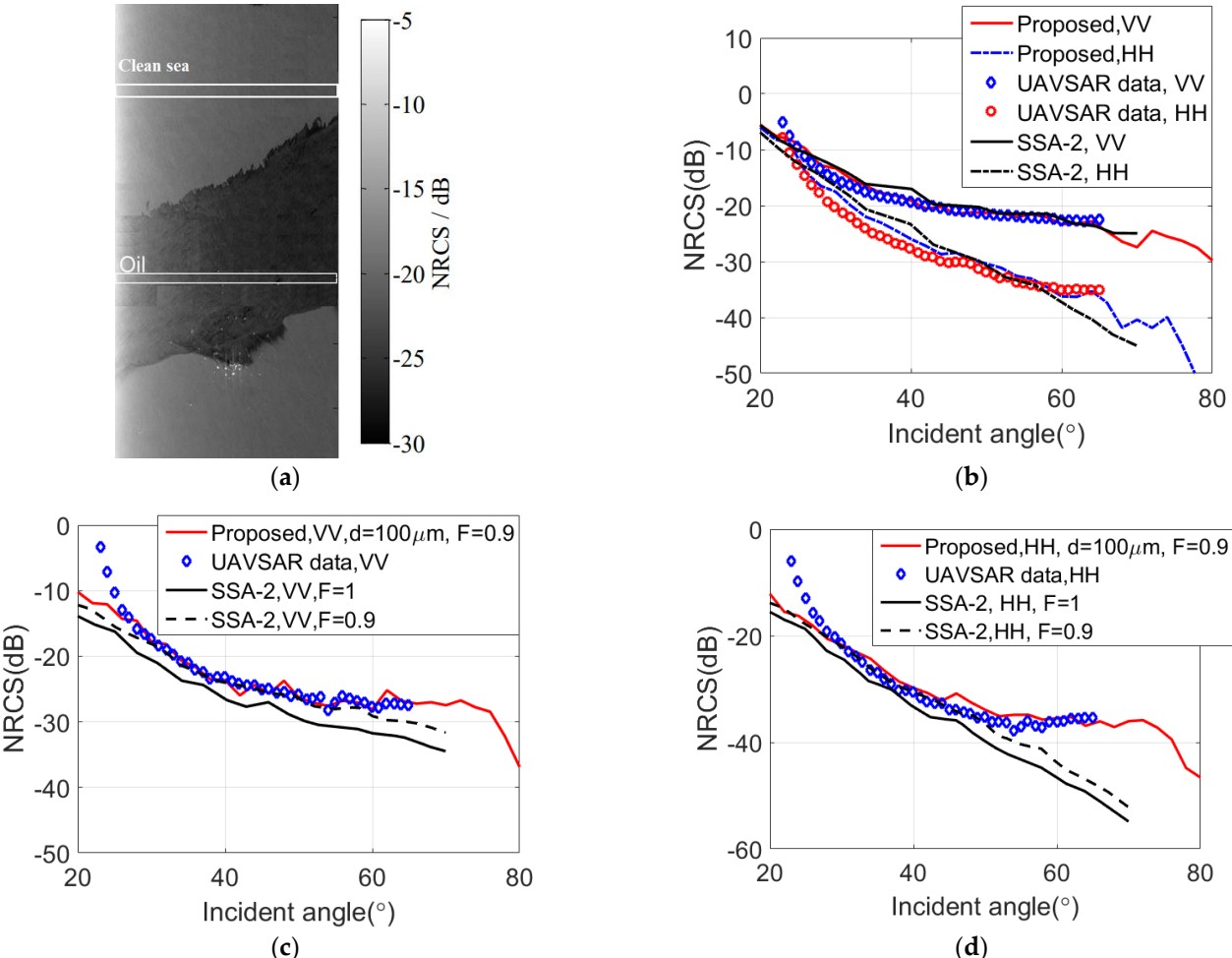

**Figure 6.** (**a**) The UAVSAR measured data, the comparison of the simulated NRCS, the SSA-2 results simulated by Zheng et al. [26], and the measured result; (**b**) Clean sea, VV, and HH; (**c**) Oil spill-covered sea and VV; (**d**) oil spill-covered sea and HH.

**Table 2.** The information of the measured data (see Refs. [42,43]).

| | Type | $f$ (GHz) | Data ID | Time | Polarization | $\theta_i$ (deg) | $U_{10}$ (m/s) | $\varphi_w$ (deg) |
|---|---|---|---|---|---|---|---|---|
| Figure 6 | UAVSAR | 1.2575 | 14010 | 20:42 UTC 17 May 2010 | VV/HH | 22–65 | 2.5–5 | 115–126 |
| Figure 7 | NASA/MSC | 13.3 | Mission 135/ Mission 156 | 21:26 GMT 16 March 1970/ February 1971 | VV | 25–50 | 9.26/ 2.5~3.08 | 110–135 |

The simulation parameters of Figure 6 are as follows: $f$ = 1.2575 GHz and $U_{10}$ = 4 m/s. Equation (6) shows that the relative dielectric parameters of the oil spill is $\varepsilon_{oil}$ = (2.96, 0.04) (mixture of 80% oil and 20% water), the seawater is $\varepsilon_{sea}$ = (72.6, 69), and $\varphi_w$ = 120° expresses the wind direction. The calculation size of the sea surface is 300 m × 300 m, and the interval sample is 1 m × 1 m.

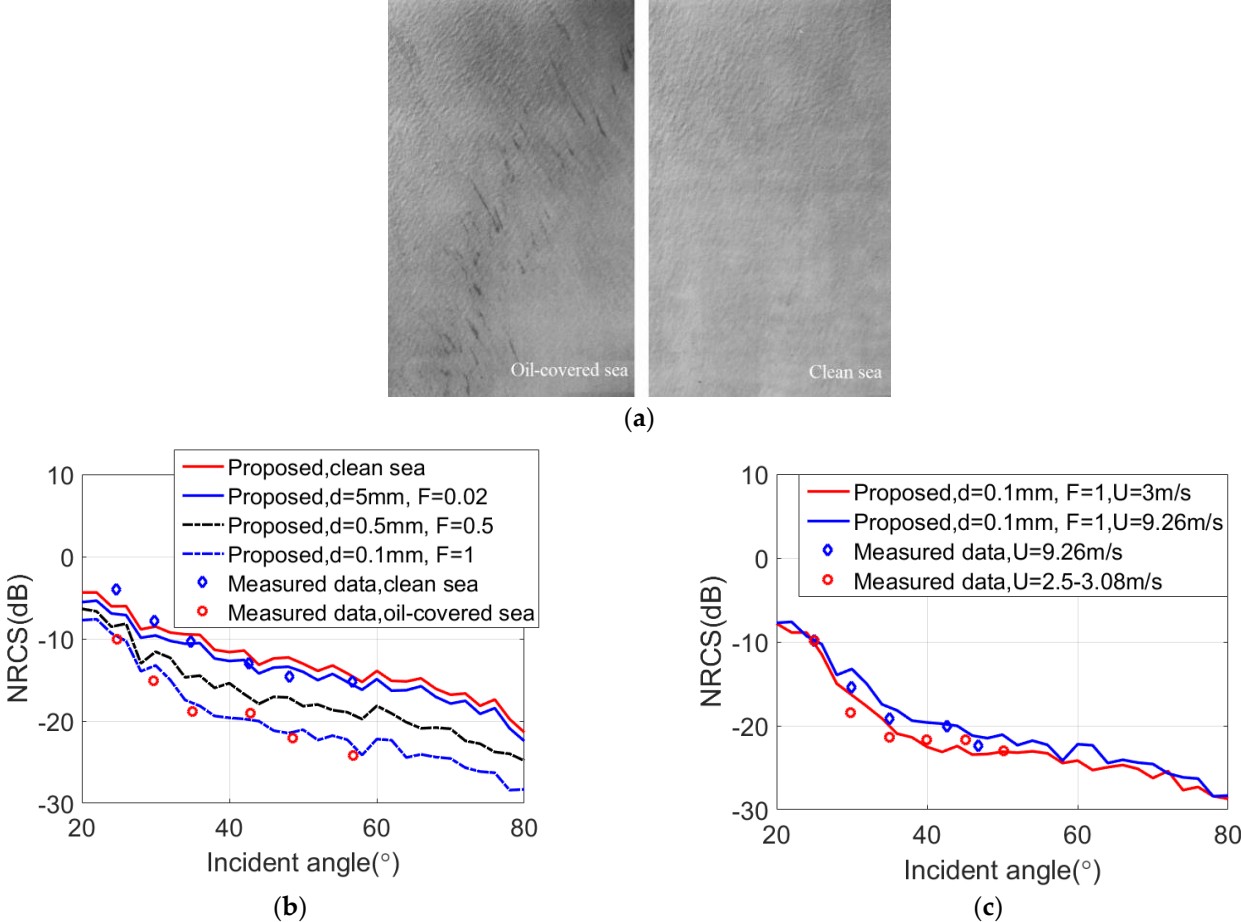

**Figure 7.** (**a**) The NASA/MSC photograph of the oil–covered sea surface area and clean sea; (**b**) Comparison of experimental data and simulation results; (**c**) Comparison of experimental data and simulation results for different wind speeds.

Figure 6b displays the NRCS of clean sea. Since there is no oil film in the clean sea, it is not necessary to involve the oil spill layer in the calculation. In Figure 6, the simulation results align well with the measured data and SSA-2 results. Figure 6c shows the EM scattering from an oil spill-covered sea surface area. It shows that the sea surface is covered by most of the oil spill, almost 90% to 100%. In the simulation, the oil spill thickness is set as $d$ = 0.1 mm (taken from [44]), and the oil spill coverage is set as 90%. The oil spill layer belongs to the oil–water mixture, and the mixing ratio is 80% oil to 20% water (taken from [45]). Under the premise that oil film covers 90% of the surface area, the simulation results align well with the measured data, which is better than the SSA-2 model results. The reason is that the simulation results of SSA–2 do not involve the EM scattering calculation of the oil spill layer; only the large–scale waves are considered. The scattering coefficient of the oil spill layer occupies an important component in the whole scattering field when the incident angle is greater than 40°. Therefore, the proposed model can well reflect the EM scattering characteristics in the actual scene. The effectiveness of the proposed model is proved.

Figure 7 shows the NRCS of the measured data [43] from clean sea and oil-covered sea surface for VV polarization. The specific information of measured data is recorded in Table 2. Figure 7a shows the photograph of the area with oil–covered sea surface and clean sea. The simulation parameters are: $f$ = 13.3 GHz, $U_{10}$ = 9.26 m/s, $\varepsilon_{oil}$ = (2.93, 0.06) (mixture of 80% oil and 20% water), $\varepsilon_{sea}$ = (48, 38.8), and $\varphi_w$ = 120°. The calculation size of the sea surface is 300 m × 300 m, and the interval sample is 1 m × 1 m. Figure 7 illustrates the

simulation results are the statistical average results of 100 samples under the conditions of VV polarization.

Figure 7b shows the EM scattering at VV polarization. As shown in Figure 6b, the simulation results for clean sea align well with SSA-II results and measured data. The changes of EM scattering during the dynamic process from the beginning of the oil spill to the diffusion are also simulated. When the oil spill thickness is $d = 5$ mm and the oil spill covers 2% of the sea area, the EM scattering intensity is smaller than that of the clean sea. As the oil spill thickness decreases, the oil spill coverage increases until it reaches $d = 0.1$ mm and $F = 1$. The simulation results are more aligned with the experimental data. (It took more than a month from oil leakage to measurement, so the parameter settings of oil spill thickness and oil spill coverage rate for this simulation are in line with the actual scene. See [39,43] for details). The main reason for the variation of EM scattering with the change of oil spill thickness and oil spill coverage is as follows: When the oil spreads out, the oil spill coverage increases, the damping effect is enhanced, the surface slope is reduced, and the sea surface becomes smooth. Therefore, the corresponding EM scattering is reduced compared with that of the clean sea.

Figure 7c shows EM scattering for different wind speeds. The comparisons shown in the figure indicate that in the presence of oil, the EM scattering feature for an ocean with a high sea state is similar to the ocean with low surface wind speed. Meanwhile, the simulation results align well with the experimental data. The reason for this phenomenon is that the larger the oil spill coverage, the stronger the damping effect. However, when the wind speed increases, although the oil spill has a damping effect on the sea surface, the high sea state will enhance the interaction between waves, and surface slope will also increase. At this moment, the damping effect of the oil spill is weakened. Therefore, for the EM scattering feature under high wind speed is similar to the situation of EM scattering at low wind speed.

### 4.2. Comparison of Hydrodynamic and Tilt Modulation

Hydrodynamic effect and tilt modulation exist in the EM scattering of oil spill–covered sea. The EM scattering analysis performed in Figure 8 studies the effects of two modulations from the oil spill-covered sea.

Figure 8a is the EM scattering simulation of grid elements in clean sea and oil spill-covered sea, where $f = 5$ GHz and $U_{10} = 3$ m/s. The small facet size is 1 m × 1 m, $\theta_n$ is the tilt of the small facet, and $\varphi_n$ is the azimuth angle. Based on the dynamic oil spill process, the oil spill changes from heavy oil to thin oil, and the oil spill thickness is considered to be reduced from 5 mm to 0.1 mm in the simulation. When the small facet is covered with an oil spill, the damping effect will lead to a smaller $\theta_n$ and a smaller RMS height of small-scale waves. The damping effect will be stronger with the increase of the oil spill thickness. This is consistent with [46].

Figure 8b shows the influence of $\theta_n$ for the scattering coefficient of the small facet at different oil spill thicknesses for VV and HH, where $\varphi_n = 190°$. Figure 8c displays the influence of $\varphi_n$ on the backscattering coefficient, where $\theta_n = 30°$. The simulation results of Figure 8 can draw the following conclusions: (1) Both hydrodynamic and tilt modulations contribute to EM scattering of the oil spill-covered facet. (2) As $\varphi_n = 190°$, NRCS decreases steadily when $\theta_n$ increases. The larger the oil spill thickness, the greater the damping effect on small-scale waves, and the smaller the NRCS of the small facet caused by an oil spill. (3) When $\theta_n = 30°$, the scattering intensity is distributed in cosine with the increase of $\varphi_n$. (4) The scattering intensity of the small facet for VV is higher than for HH, which can more clearly reflect the changes brought about by the existence of an oil spill. (5) Compared with the hydrodynamic effect in the small facet, the tilt modulation is the main factor. As time goes by, the dynamic oil changes and the scattering coefficient of the small facet increases. The EM scattering of the whole observed sea surface is actually the comprehensive effect of the oil spill layer and the oil spill-covered surface area.

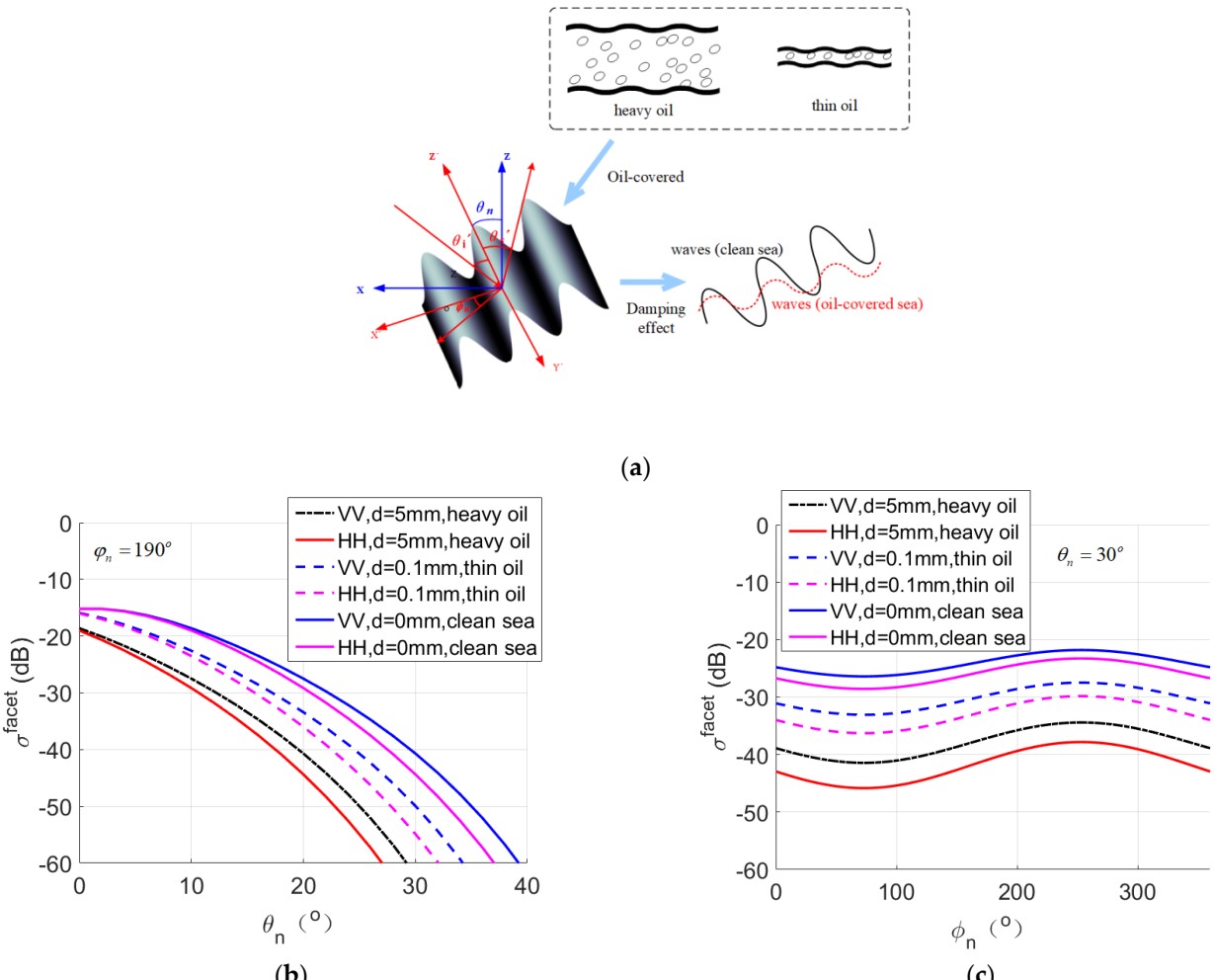

**Figure 8.** (**a**) The EM scattering simulation of grid elements in clean sea and oil spill–covered sea surface; (**b**) Influence of $\theta_n$ on backscattering coefficient; (**c**) Influence of $\varphi_n$ on backscattering coefficient.

## 5. Conclusions

In the manuscript, the EM scattering mechanisms of dynamic oil spill–covered sea area are studied. The two-scale model and oil spill diffusion theory are combined to generate the dynamic geometric model of an oil spill–covered sea surface area. The thinner the oil spill layer, the stronger the damping effect. The modified two-scale model and the multilayer medium scattering model are adopted for calculating the EM scattering of oil spill–covered sea areas. Compared with UAVSAR and NASA's actual experimental oil spill data, the proposed model has a better calculation accuracy than SSA–2. In addition, the EM scattering in terms of different frequencies, wind speed, oil film thickness, and oil coverage area are also simulated. The EM scattering simulation of grid elements in clean sea and oil spill–covered sea surface concluded that the contribution of tilt modulation is greater than the hydrodynamic effect for dynamic oil spill diffusion. The proposed model provides a new theoretical reference for oil remote sensing monitoring. However, the purpose of studying the EM scattering of the oil spill–covered sea is to effectively predict the oil leakage amount and the oil film thickness through the EM simulation platform. Therefore, more theoretical research will be carried out on how to accurately estimate the oil leakage amount in future works.

**Author Contributions:** D.L. drafted the manuscript and was responsible for theoretical research and simulation; Z.Z. supervised the research and was responsible for the review of the manuscript; W.M. and Y.X. were responsible for reviewing the manuscript and verifying the data. All authors have read and agreed to the published version of the manuscript.

**Funding:** This research was supported in part by the National Natural Science Foundation of China under Grant 61871083, Grant 62031010, and Grant 61721001; in part by the Through Train Project for Ph.D. of Chongqing, China, under Grant CSTB2022BSXM-JSX0004; and in part by the Natural Science Foundation of Sichuan Province, China, under Grant 2022NSFSC1860 and Grant 2023NSFSC0258.

**Conflicts of Interest:** The authors declare no conflict of interest.

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
