# Peer review of "Investigation of Electromagnetic Scattering Mechanisms from Dynamic Oil Spill–Covered Sea Surface"

_remotesensing, doi:10.3390/rs15071777_

Round 1

Reviewer 1 Report

The scope of this paper is perfectly suitable for Remote Sensing. In this work, the authors investigate the mechanisms of electromagnetic scattering from dynamic oil spill-covered sea surfaces by using a two-scale model and oil spill diffusion theory. The simulation results clearly explain what mechanism is responsible for the RCS modulations in process of dynamic oil spill. The numerical results have been compared with measurements and show good agreement. I think it is innovative to compare the effects of hydrodynamic and tilt modulation in a particular application, and the work in this paper makes a great reference to oil remote sensing monitoring.

The paper is well written. The research of this paper is a very contribution to the remote sensing community. I recommend publication subject to some minor comments as follows:

1. Figure 3 shows the RMS slopes in x direction, it would be helpful if the authors could add the RMS slopes in y direction as well.

2. The legend of the curve in Fig. 8(b) is not correct, it should be “d=0.1mm”, rather than “d=0.1mmm”. And the explanations of Fig. 8 should be more detailed.

3. In Fig. 4, the coordinate systems should be clearly shown in one figure.

4. Several references are given in an incorrect form, please correct them.

Reviewer 2 Report

This manuscript studied the electromagnetic scattering mechanism of dynamic oil spill-covered sea surface. According to the theory of oil film diffusion, combined with the oil spill volume, a three-dimensional (3D) geometric model of dynamic oil spill-covered sea surface is established. The experiments show that the proposed model has better performance. This manuscript fails to highlight the innovation very well.The recommendations are as follows.

1.In the introduction, the authors describe a variety of fine geometric models in the EM scattering from sea surface covered by dynamic oil spill. However, there is no presentation of the scattering mechanism of electromagnetic waves based on oil spills, such as Bragg scattering.

2.The meaning of the coordinates an the source of the relevant data in Figure 1 should be stated.

3.In the 3D geometric model of oil spill-covered sea surface, some symbols have unclear meanings. For example, line 109 on page 3, please check if τ appears in the formula 2-4 and formula 10, what IFFT2 means. 

4.In the sea surface, how to define e large-scale waves? 

5.the authors need to focus on innovation. The conclusions should be presented in a more general form, emphasizing what dataset was used, how it broadened our knowledge, and whether it made a new contribution to the methodology of studying detection. It is also worth writing about future research directions.

6.There are some minor problems. For example, spelling errors exist in line 169 on page 5 and the title of Figure 7.

Reviewer 3 Report

The authors have studied the EM scattering mechanisms of dynamic oil spill-covered sea. The NRCS coefficient is obtained and compared with the measured data. Specific effects of hydrodynamic and tilt modulations are discussed. The manuscript is not acceptable in its current form, and it is necessary to clarify some items for the readers:

1)     The main contribution of the work is not clear in the introduction.

2)     How are the values in Table 1 obtained?

3)     The words “dynamic” and “mechanism” are meaningless as keywords.

4)     Eqs. (7)-(9) and (15)-(17) need references.

5)     w is used in this paper in what sense and what values can it take?

6)     In the caption of Fig. 1, introduce all the parameters on the figure (K, S, K_min, K_cut, K_Bragg).

7)     The explanations related to Figs. 3(a) and 3(b) are not clear.

8)     Numbers (2) and (4) have disappeared among the circles in Fig. 5.

9)     It is not clear how the Eqs. (18)-(20) were obtained.

10) At the end of the caption of Table 2, the reference should be declared (probably [43, 44]).

11) In Section 5, provide a summary of the results (preferably quantitative).

12) Some writing comments:

-        Lines 19, 71, 126, …: Do not start the sentence with “And”.

-        Lines 34, 42, …: What is meant by PM?

-        Many parts of the text should not be expressed in a new paragraph; for example, lines 95, 107, 109, …

-        Please use a comma or dot at the end of equations.

-        Parameters must be written in italics, e.g. line 109, line 180, Fig. 3, …

-        What does * mean in line 130?

-        Line 191: 300m*300m → 300 m × 300 m

-        RMS should be introduced in full in the first use.

-        Align Eqs. (11) and (12).

-        Line 205: surface(The → surface (the

-        Line 209: surface.As → surface. As

Round 2

Reviewer 3 Report

The authors have addressed major comments. However, there are still some minor/writing issues:

1)     Some formulas still require references. The corresponding reference must be provided before or after the formula.

2)     Please observe the space between the characters; for example line 252, …

3)     Many parts of the text should not be expressed in a new paragraph; for example, lines 104, 112, 143, 162, …

4)     Lines 68 and 124: Do not start the sentence with “And”.

5)     Parameters must be written in italics, e.g. line 112, Figs. 1 and 3, …

6)     Index w is used in this paper in what sense and what values can it take?

7)     What does * mean in line 128?
